# Chemical Recycling of Mixed Polyolefin Post-Consumer Plastic Waste Sorting Residues (MPO323)—Auto-Catalytic Reforming and Decontamination with Pyrolysis Char as an Active Material

**DOI:** 10.3390/polym16182567

**Published:** 2024-09-11

**Authors:** Tobias Rieger, Martin Nieberl, Volodymyr Palchyk, Pujan Shah, Thomas Fehn, Alexander Hofmann, Matthias Franke

**Affiliations:** Fraunhofer Institute for Environmental, Safety and Energy Technology Umsicht, Institute Branch Sulzbach-Rosenberg, An der Maxhütte 1, 92237 Sulzbach-Rosenberg, Germany; martin.nieberl@umsicht.fraunhofer.de (M.N.); volodymyr.palchyk@umsicht.fraunhofer.de (V.P.); thomas.fehn@umsicht.fraunhofer.de (T.F.); matthias.franke@umsicht.fraunhofer.de (M.F.)

**Keywords:** mixed plastic waste, chemical recycling, pyrolysis, recovery of aromatics, oil upgrading, dehalogenation, reforming, decontamination

## Abstract

Mixed plastic packaging waste sorting residue (MPO323) was treated by thermal pyrolysis to utilize pyrolysis oil and char. The pyrolysis oil was found to contain aromatic and aliphatic hydrocarbons. The chlorine and bromine contents were as high as 40,000 mg/kg and 200 mg/kg, respectively. Additionally, other elements like sulfur, phosphorous, iron, aluminum, and lead were detected, which can be interpreted as impurities relating to the utilization of oils for chemical recycling. The pyrolysis char showed high contents of potentially active species like silicon, calcium, aluminum, iron, and others. To enhance the content of aromatic hydrocarbons and to reduce the level of contaminants, pyrolysis oil was reformed with the corresponding pyrolysis char to act as an active material in a fixed bed. The temperature of the reactor and the flow rate of the pyrolysis oil feed were varied to gain insights on the cracking and reforming reactions, as well as on performance with regard to decontamination.

## 1. Introduction

Monocyclic aromatic compounds such as benzene, toluene, ethylbenzene, and xylene (BTEX) are basic chemicals used for making a wide range of intermediates in the fields of packaging, pharmaceuticals, adhesives, coatings, pesticides, automotive technology, and others [1]. Aromatic compounds are normally produced in the steam cracking process, where naphtha, liquified petroleum gas, or gas oil is vaporized, mixed with water vapor, and heated to up to 850 °C in the presence of catalysts to achieve pyrolytic cracking and conversion of saturated hydrocarbons into C2- to C4-olefines and aromatic compounds [2]. From the product mixture, aromatic compounds such as BTEX are isolated via extraction or extractive distillation [2]. An alternative recycling-based feedstock for steam cracking can be obtained by chemical recycling of plastic wastes.

Mechanical recycling of plastic wastes, which includes sorting and reprocessing into new plastic products, is the preferred recycling option for plastic wastes. However, contaminants, intermixture with other plastics and/or materials, and thermal-mechanical degradation pose limitations to the recycling of plastic wastes via mechanical recycling [3]. As an example, MPO323 is a fraction produced via near infrared (NIR) sorting of lightweight packaging waste [4]. Despite state-of-the-art NIR sorting, MPO323 comprises only ≥85.0 wt% of the target plastics PP and PE; the remainder is impurities such as paper, cardboard, carton (<5.0 wt%), plastics such as PET and PS (<7.5 wt%) or PVC (<0.5 wt%), and other impurities such as rubbers or food residues (<3.0 wt%) [5]. These impurities impede the reprocessing of NIR-sorted PP and PE wastes.

Chemical recycling of plastic waste offers the advantage of complementing established mechanical recycling and of being able to produce high-quality recyclates from previously non-recyclable feedstocks. By breaking down the polymer chains into smaller molecular fragments, chemical recycling processes make it possible to further separate both contaminants and the various additives contained in plastic products. Pyrolysis is a simple technology, suitable to treat highly heterogenous mixtures of plastics [3]. The pyrolysis oil from plastic waste pyrolysis can be used in steam crackers to produce aromatic hydrocarbons and olefins, substituting fossil-based naphtha. Most pyrolysis oils, however, contain relevant amounts of heteroatom contaminants such as nitrogen, oxygen, halogens, and metals that stem from heteroatom-containing polymers such as PVC, PET, or PUR, as well as additives such as flame retardants, CaCO_3_, SiO_x_, etc. [6]. These contaminants are undesired in petrochemical processes like steam cracking and BTEX isolation, as they cause catalyst poisoning as well as corrosion and fouling of process equipment [7]. In general, crude oils from the pyrolysis of mixed packaging plastics exceed the maximum limits of steam crackers for contaminants such as chlorine and bromine [6]. For pyrolysis oils to be applicable in petrochemical processes, the level of contaminants needs to be reduced and fall below process specific maximum thresholds [6].

To minimize contaminant concentrations in pyrolysis oils, researchers investigated approaches for pre-treatment of plastic wastes, post-treatment of pyrolysis oils, and online approaches applied during pyrolysis. These approaches have been reviewed in [6,7,8].

Pre-treatment aims at a reduced input of heteroatom contaminants into the pyrolysis process, and consequently into the pyrolysis products. Pre-treatment approaches encompass washing the plastics wastes [9], sorting out unwanted plastics such as PVC, e.g., via triboelectrostatic separation [10] and froth flotation [11], hydrothermal dehalogenation [12], and mechanochemical dehalogenation [13]. These approaches allow the reduction of heteroatom contamination in pyrolysis oils. However, they do not achieve complete removal of chlorine and other contaminants. Therefore, online and post-treatment approaches are also needed to produce pyrolysis oil with sufficiently low heteroatom contamination levels.

Post-treatment approaches aim to remove heteroatom contaminants from pyrolysis oil. Via filtration and fractional distillation of pyrolysis oil, Rieger et al. produced distillates with bromine and chlorine contents 99% lower than in the original oil [14]. Another post-treatment approach to remove heteroatoms from pyrolysis oils is catalytic hydrotreatment, which was investigated in [15,16]. Even though hydrotreatment is a promising way to remove traces of contaminants from pyrolysis oil, a too high heteroatom load is likely to occupy active catalyst sites, leading to decreased efficiency [7], making it necessary to reduce the content of heteroatoms in oils prior to hydrotreatment. Kusenberg et al. conclude that hydrotreatment processes should be designed specifically for plastic waste pyrolysis oils, considering their unique composition and concentration of contaminants [17].

Online approaches encompass stepwise pyrolysis and the use of dehalogenation agents in the pyrolysis reactor (in situ) or in a separate vessel (ex situ). Stepwise pyrolysis is particularly well suited to remove chlorine from PVC, since thermal degradation of PVC is a two-staged decomposition process in which maximum HCl is released at 320 °C and maximum hydrocarbons at 480 °C [18]. With stepwise pyrolysis, Park et al. converted a mixture of waste LDPE, PP, and PVC into a pyrolysis oil that exhibited 87% less chlorine than oil from single-step pyrolysis [19]. Another frequently investigated approach is the in situ or ex situ utilization of alkaline, acidic, or metallic dehalogenation agents that accomplish C-X bond cleavage, HX formation, and/or X-fixation [7]. Cho et al. reduced the chlorine content in the oil from mixed plastic waste pyrolysis from 502 ppm to 50 ppm using Ca(OH)_2_, and to 58 ppm by using CaO [20]. Miskolczi et al. reported the chlorine and bromine contents of oils from the pyrolysis of MSW with Y-zeolite, β-zeolite MoO_3_, Ni-Mo catalyst, HZSM-5, and Al(OH)_3_. The best results were obtained with Al(OH)_3_, where chlorine and bromine contents were 1954 ppm and 71 ppm, respectively; considerably lower compared to 3041 ppm chlorine and 914 ppm bromine in the pyrolysis oil when no agent was used at all [21]. The online use of dehalogenation agents can lead to substantially reduced contents of chlorine and bromine in pyrolysis oils and is therefore helpful in producing contaminant-free pyrolysis oils. Kusenberg et al. remark that, in the case of sorbents, their limited capacity requires frequent exchange and regeneration [6]. Agents that exhibit catalytic effects also may be deactivated quickly due to heteroatom contamination. These challenges can be overcome by utilizing pyrolysis char, the solid residue from pyrolysis, as an inexpensive, continuously arising dehalogenation agent. Areeprasert and Khaobang investigated Y-zeolite, ZSM-5, and the chars from electronic waste pyrolysis and from biomass pyrolysis, respectively, for their capability to produce bromine free oils from ABS/PC pyrolysis. The authors found that iron in the pyrolysis char has a debromination effect and were able to reduce the bromine content by 91% using the char from electronic waste pyrolysis additionally loaded with iron oxide [22]. These results indicate that pyrolysis char may be an effective dehalogenation agent to produce pyrolysis oils exhibiting low contents of halogens and possibly other heteroatom contaminants. However, literature on the use of pyrolysis char as a dehalogenation agent or, more broadly, as a decontamination agent is very limited.

Pyrolysis oils from mixed plastic wastes also contain a wide range of aromatic compounds [17,23]. The aromatic compounds in the pyrolysis oils constitute a weakness on the one hand and a chance for high-quality recycling of plastic wastes on the other hand. Aromatic compounds impair the steam cracking process, as they lead to increased coke formation and fouling in heat exchangers [24,25]. Highly aromatic pyrolysis oils, however, can be directly applied in downstream process such as BTEX isolation or other petrochemical processes, thus making energy-intensive steam cracking dispensable and allowing for higher-quality recycling of plastic wastes. To tap this chance and produce highly aromatic pyrolysis oils from plastic wastes, researchers investigated in situ and ex situ catalytic reforming during pyrolysis. Frequently investigated catalysts are zeolites [26,27] loaded with metals [28] and of different pore sizes and acidities [29]. Zhang et al. achieved a selectivity of 90.7% monocyclic aromatic compounds and 77.6% BTEX using an HZSM-5 zeolite with a SiO_2_/Al_2_O_3_ ratio of 25 and loading of 3 wt% gallium and 2 wt% phosphorus for ex situ catalytic reforming of vaporous products from LDPE pyrolysis [30]. However, conventional catalysts show reduced activity, including deactivation, especially during treatment of mixed plastic waste [1].

Similarly to dehalogenation agents, these drawbacks can be overcome by using inexpensive catalysts that can be easily replaced once their activity decreases. Pyrolysis char was proposed as such a catalyst and investigated by several researchers. Sun et al. investigated sewage sludge char for selective production of aromatics in the pyrolysis of waste mixed plastics. The highest yield of 75.3% monocyclic aromatic compounds was obtained at a 600 °C catalytic temperature and a 1 sec residence time. The authors concluded that ash components in the catalyst increase the aromatization degree, with acid sites (mainly aluminum phosphate) and dehydrogenation active sites (mainly phosphates, sulfides, etc.) catalyzing direct dehydrogenation, a hydrogen transfer reaction, and Diels–Alder reactions thereby promoting the formation of aromatic hydrocarbons [31]. Qian et al. prepared catalysts for the pyrolysis of LDPE, PP, PS, and PET by impregnating industrial organic solid waste, consisting of plastics, fabrics, paper, and wood, with H_3_PO_4_ or ZnCl_2_ and pyrolyzed the impregnated waste. The catalysts exhibited high specific surface areas greater than 600 m^2^/g, and the phosphorus- and zinc-involved acid sites promote C-C cracking and aromatization of polyolefins [32]. In [33], catalysts for pyrolysis of plastic wastes were prepared by impregnating wood chips with KOH, ZnCl_2_, and H_3_PO_4_ and pyrolyzing them. KOH-activated biochar promoted hydrogen transfer processes, increasing the yield of aromatic compounds. Treatment with ZnCl_2_ and H_3_PO_4_ led to Lewis/Bronsted acid sites on the char, promoting dehydrogenation processes, hydrogen transfer reactions, and the Diels–Alder reaction to convert alkenes into aromatic compounds. Fan et al. used char from pyrolysis of municipal solid waste activated with Na_2_CO_3_, Zn(NO_3_)_2_⋅6 H_2_O, and ZnCl_2_ as a catalyst for the pyrolysis of municipal solid waste. The highest selectivity for monocyclic aromatic compounds, with 47 area% and for H_2_ in the pyrolysis gas, was exhibited by ZnCl_2_ activated char. The Zn [OH]^+^ species and high L-acid contents enhance hydrogen transfer during the aromatization process [34].

To the best of our knowledge, no other pyrolysis chars have been investigated as catalysts for the conversion of pyrolysis vapors from plastic pyrolysis into monocyclic aromatic compounds.

Against this background, this study investigates an alternative approach of producing highly aromatic pyrolysis oil of low heteroatom contamination, which does not rely on steam cracking as a final utilization step. This study uses MPO323 as feedstock for pyrolysis and the char of MPO323 pyrolysis as both a decontamination agent and a catalyst for aromatization. This work aims to add to the limited knowledge in this field by experimentally investigating the decontamination and catalytic effects of char from MPO323 pyrolysis.

## 2. Materials and Methods

### 2.1. Mixed Polyolefin Post-Consumer Plastic Waste (MPO323)

Real municipal solid plastic waste (MSPW) sorting residues from the German recycling system was provided by one of the leading German public waste disposal companies. A standardized fraction named mixed polyolefins (MPO323) was used for the present work. The material was provided and processed in the form of pellet-like agglomerates with a diameter of approximately 6 mm and a length of approx. 12 mm. Due to repeated transfer for transporting and storing reasons, the length of the agglomerate particles varied greatly. The material is shown in Figure 1.

### 2.2. Thermo-Chemical Conversion

Four pyrolysis experiments were conducted in Fraunhofer UMSICHT’s semi-continuous pyrolysis system with a capacity of 0.6 kg/h. The system is shown in Figure 2. It consists of an inert gas cylinder (1), hand valves (2, 3), a collection container for pyrolysis char (4), a pyrolysis reactor (5), a transition pipe (6), a tar filter (7), two spiral coolers (8, 9), a three-way valve (10), a wash bottle with NaOH (11), a wash bottle with water (12), a three-way valve (13), an activated carbon filter (14) with discharge to the ambient air, a motor-driven stirrer (15), a feed system (16, 17, 18), and an electrical heating jacket around the reactor (5).

The reactor (5) has a volume of 5 L. The pipe (6) and the tar filter (7) are 2-inch standard piping transitioning to 16 mm pipes with 1.5 mm wall thickness in the condensation system (8, 9). Silicon tubes were used to connect the washing system (11, 12) and gas analysis system. All stainless steel components are made from 1.4575 stainless steel. The feed system includes a dedicated nitrogen purge unit to ensure displacement of ambient air from the feedstock before each feed addition. The nitrogen flow of the purge unit is controlled by a volume flow rate controller and set to 0.5 L/min for purging. The nitrogen flow through the reactor system during any experiment is ensured by another volume flow rate controller and is set to 2 L/min. The stirrer (15) is operated with a pneumatic motor and set to approximately 60 RPM for all experiments. The wash bottles (11) and (12) are filled with 800 mL 33% (*w*/*w*) sodium hydroxide solution and 800 mL distilled water, respectively.

Prior to any experiment, the setup was pressurized to at least 0.5 bar to test the system for pressure stability and leakage. Pressure tests were considered successful when the change in pressure was less than 5 mbar per second.

The system is operated in semi-batch mode, i.e., a batch of desired quantity of feedstock was filled into the feed system (16), purged with nitrogen, and fed into the reactor through the locking system (17). Each batch is processed for 20 min. During this time, the feed system is refilled with material and purged with nitrogen. After 20 min have expired, the next batch was fed into the reactor. This procedure was repeated until the end of the experiment. To prepare pyrolysis oil and char for the present investigation, a total of 6.214 kg of MPO323 was treated by the conversion process described above within four experiments at a reactor temperature of 520 ± 10 °C.

### 2.3. Auto-Catalytic Reforming Process

Laboratory-scale reforming experiments were performed to obtain insights into the catalytical activity of compounds contained in pyrolysis chars regarding the aromatization and decontamination processes. For this purpose, a laboratory setup with a fixed-bed reactor tube and a subsequent condenser was developed. A syringe pump fed pyrolysis oil to the reactor tube, in which 5 g pyrolysis char from MPO323 (produced by the method described in Section 2.2) was placed in the form of a packed bed. The pyrolysis oil feed rate was varied between 0.05 mL/min and 0.45 mL/min. The wall of the reactor tube was heated by an electrical heater to between 500 °C and 700 °C. The oil entering the reactor evaporated and passed the char bed to promote the aromatization and decontamination processes. Subsequently, the gas was condensed in a water-cooled condenser and collected for analysis. This setup is described in the following text.

Figure 3 shows a process flow diagram of the laboratory setup of the catalytical reforming setup. A quartz glass cylinder (1) with a wall thickness of 1 mm and a diameter of 10 mm was used as a reactor. The pyrolysis char bed (2) was placed in the center of the reactor and held in position by quartz wool. The upper end of the reactor was sealed with a PTFE-lined silicone rubber septum, through which the nitrogen was supplied (3), and a cannula with a diameter of 0.8 mm was introduced to feed pyrolysis oil (4). The pyrolysis oil was supplied via a syringe pump (5), which generated a constant oil flow with a volume flow rate of 0.05–0.45 mL/min. A tubular heater was placed around the reactor, and a thermocouple was attached to the center of the heating zone of the reactor to control the temperature. Directly below the reactor a water-cooled Liebig condenser (6) was installed. The resulting product was collected in a receiving flask (7) located below the cooler. The excess gas flowed out of the receiving flask into the flue and was then treated by an activated carbon filter. The receiving flask was chilled in an ice water bath.

Five experiments with one repetition each were conducted as listed in Table 1 within a 2^2^ full factorial design of experiments to gain statistical data. The obtained data were evaluated by using a response surface model (RSM) and ordinary least square regression (OLS). The order of experiments was randomized to minimize systematic errors. The target function was of the following type:Y = X1 + X2 × T + X3 × F + X4 T/F(1)
where Y is the target value, T is the temperature, F is the volume flow rate, and X1, X2, X3, and X4 are the corresponding constants or factors. The model has been used to generate statistical data and gain insights into whether temperature or flow rate affect the selected target value. As a result of limited data points, the model does not usually describe the real behavior of the correlations, but gives statistical validation for certain qualitative trends and behaviors.

### 2.4. Analytical Methods

#### 2.4.1. Gas Chromatography-Mass Spectrometry (GC-MS) Analysis

The composition of oil was analyzed on a gas chromatograph (GC) coupled with a mass spectrometer (MS) (Shimadzu, Kyoto, Japan; GCMS-QP2020). The GC was equipped with a 30 m nonpolar 0.25 mm inner diameter (i.d.), 0.25 µm film thickness DB-5ms and a 2.5 m middle polar 0.15 mm i.d., 0.15 µm film thickness VF-17ms column set from Agilent Technologies. Helium with 5.0 purity was used as carrier gas for all experiments. The injection volume was set to 1 µL, split 1:20. Dilution: 1 mg of sample in 1 mL of dichloromethane (DCM). The measurements were performed at a constant linear carrier gas velocity of 40 cm/min. The temperature of the GC oven was programmed as follows: 40 °C, 3 min hold to 320 °C, 3 min hold at 10 °C/min. The temperatures of the injector, MS-interface, and MS were set to 250, 280, and 200 °C, respectively. The quadrupole MS detector operated at a scan speed of 5000 Hz using a mass range of 35–500 *m*/*z*. The solvent cut time was 3 min, and the MS start 3.2 min. The total analysis time was 34 min. The 25 biggest peaks were integrated. BTEX, styrene, α-methyl styrene, phenol, cresols, and naphthalene were identified using standard solutions of pure chemicals. The NIST-17 Mass Spectral Library was used to identify other substances with a similarity index more than 70%. For some aliphatic hydrocarbons, exact identification was not possible; therefore, the number of carbon atoms was provided. The assignments were made using C7-C30 standards (Supelco, UK). The proportion of each substance in the sample is given in area%.

#### 2.4.2. Elemental Analysis (CHNS)

Determination of CHNS composition was performed on Elementar vario macro cube (Elementar Anlagensysteme GmbH, Langenselbold, Germany). Measurements were conducted analog to DIN 51732 [35] and performed three times.

#### 2.4.3. Inductively Coupled Plasma with Optical Emission Spectrometry (ICP-OES)

ICP measurements for oil samples were performed using a Spectro Arcos ICP-OES (SPECTRO Analytical Instruments GmbH, Kleve, Germany) in dual side on plasma mode. The ICP was equipped with a Nordermeer nebulizer, a cyclonic spray chamber, and a fixed glass torch with a 1.8 mm injector. The measurement conditions were radio frequency power 1350 W, plasma gas flow 14 L/min, nebulizer flow 0.72 L/min, and auxiliary gas flow 2 L/min. The sample aspiration rate was set to 2.0 mL/min. All liquid samples and standards were diluted 1:10 with 1-Butanol. Solid samples were digested in aqua regia based on DIN EN ISO 54321 [36]. For measurements, a cross-flow nebulizer, a Scott spray chamber, and a fixed glass torch with a 1.8 mm injector were used. The measurement conditions were RF power 1350 W, plasma gas flow 14 L/min, nebulizer flow 0.75 L/min, and auxiliary gas flow 1.3 L/min. The sample aspiration rate was set to 2.0 mL/min.

#### 2.4.4. Surface Area, Pore Size, and Pore Volume

The surface area, pore size, and pore volume were determined using a Belsorp Max 2 (Microtrac Retsch GmbH, Haan, Germany) according to DIN ISO 9277-2014-01 [37].

## 3. Results

### 3.1. Thermo-Chemical Conversion

Conventional pyrolysis of MPO323 sorting residues at 520 °C resulted in 29.81 ± 4.01 wt% pyrolysis char, 46.40 ± 3.00 wt% pyrolysis oil, and 23.79 ± 5.91 wt% pyrolysis gas, as shown in Figure 4.

On average, the liquid product contained 41.05 ± 3.76 area% of aliphatic hydrocarbons (AHs), 38.74 ± 0.38 area% of benzene, toluene, ethylbenzene, and xylene (BTEX), 51.37 ± 0.34 area% of BTEX and styrene (BTEXS), 58.96 ± 3.77 area% of monocyclic aromatic hydrocarbons (MAHs) in total, and no polycyclic aromatic hydrocarbons (PAH). Table 2 shows a representative GC-MS peak table of one measurement of the produced pyrolysis oil. Ethylbenzene, styrene, and toluene constituted the main aromatic products, and 2,4-dimethyl-1-heptene was the most present aliphatic compound.

ICP-OES analysis revealed concentrations of chlorine, sulfur, bromine, iron, phosphorus, lead, and aluminum in raw pyrolysis oils. MPO323 pyrolysis oil contained 42,887 mg/kg of chlorine, 788 mg/kg of sulfur, 195.33 mg/kg of bromine, 32.27 mg/kg of phosphorus, 28.8 mg/kg of iron, 1.22 mg/kg of aluminum, and 0.15 mg/kg of lead as shown in Table 3 with corresponding deviations.

Pyrolysis char produced from real MPO323 sorting residues contained carbon (C), hydrogen (H), nitrogen (N) and sulfur (S), as well as the elements aluminum (Al), barium (Ba), calcium (Ca), copper (Cu), chromium (Cr), iron (Fe), potassium (K), magnesium (Mg), manganese (Mn), sodium (Na), nickel (Ni), phosphorus (P), lead (Pb), sulfur (S), silicon (Si), titanium (Ti), and zinc (Zn) in concentrations above 500 mg/kg. The detailed elemental composition of the pyrolysis char is given in Table 4.

BET analysis revealed that the produced pyrolysis char comprises a specific surface area of 5.3 m^2^/g, a pore volume of 0.013 cm^3^/g, and a pore size of 10.2 nm.

### 3.2. Auto-Catalytic Reforming Process

Pyrolysis oil derived from MPO323 sorting residues was reformed over a fixed bed containing the corresponding pyrolysis char at various temperatures and flow rates of 500 °C, 600 °C, and 700 °C, and 0.05 mL/min, 0.25 mL/min, and 0.45 mL/min to obtain insights on the performance regarding the increase of aromatic compounds and the removal of different species in the liquid product. The investigated flow rates of 0.05 mL/min, 0.25 mL/min, and 0.45 mL/min can be translated to weight hourly space velocities (WHSV) of approximately 0.2 h^−1^, 1.1 h^−1^, and 2 h^−1^, respectively.

The mass balance of the auto-catalytic reforming experiments (Figure 5) shows that the liquid yield decreased with increasing temperature and decreasing flow rate. The highest yield of 89.50 ± 2.12 wt% was obtained at 500 °C with a flow rate 0.45 mL/min. The lowest yield of 30.50 ± 4.95 wt% was observed for parameters of 700 °C and 0.05 mL/min.

The composition of summarized categories aliphatic hydrocarbons (AHs), BTEX, BTEXS, monocyclic aromatic hydrocarbons (MAHs), polycyclic aromatic hydrocarbons (PAHs), and total aromatic hydrocarbons (TAHs), dependent on temperature and flow rate, are shown in Figure 6. As described in Section 3.1, the untreated pyrolysis oil already contained roughly 41 area% of AH and 59 area% of TAH.

The opposite trend was observed for the content of TAH in the liquid product. With increasing temperatures and decreasing flow rates, the amount of TAH was increased. Fully aromatic liquid products (99.50 ± 0.73 area% at 0.45 mL/min and 99.80 ± 0.25 area% at 0.05 mL/min) were produced at 700 °C. Liquid yields dropped to 40.00 ± 5.66 wt% and 30.50 ± 4.95 wt% at the named conditions, respectively. Only traces of aliphatic compounds were detected in the products obtained at 700 °C. PAHs were not formed in experiments at 500 °C.

BTEX reached the highest content in the product of 66.19 ± 0.54 area% at 600 °C and 0.25 mL/min. The category BTEXS was found to show the highest proportions in the liquid product at 600 °C and 0.25 mL/min (77.01 ± 0.12 area%) and 700 °C and 0.45 mL/min (80.45 ± 2.27 area%). The product at 700 °C and 0.45 mL/min also comprised the highest proportion of styrene, with roughly 18 area%.

Concentrations of chlorine, sulfur, iron, phosphorus, bromine, lead, and aluminum were measured in the pyrolysis oil of MPO323 sorting residues and had concentrations of 42,887.00 ± 747.23 mg/kg, 788.00 ± 3.46 mg/kg, 1.22 ± 0.43 mg/kg, 28.80 ± 4.92 mg/kg, 32.27 ± 0.15 mg/kg, 195.33 ± 3.51 mg/kg, and 0.15 ± 0.06 mg/kg, respectively.

The concentration of chlorine in the liquid product of the reforming process could be reduced to 141.00 ± 5.66 mg/kg at 500 °C and 0.05 mL/min, which corresponds to a reduction of 99.67% in relation to the initial pyrolysis oil. The least reduction of chlorine, to 4919.00 ± 4221.43 mg/kg (88.53%), was obtained at 700 °C and 0.45 mL/min, comprising a high standard deviation. It was found that lower temperatures and lower flow rates improved the removal efficiency of chlorine. The flow rate or WHSV had a significantly higher impact on the chlorine removal efficiency.

The content of sulfur was reduced at 600 °C by 13.83%. At 500 °C better reductions of sulfur of 27.92 and 47.08% were achieved with flow rates of 0.45 mL/min and 0.05 mL/min, respectively. This indicates that the lower flow rates and temperatures promote the removal of sulfur to a certain extent. The lowest sulfur concentration in the product was 417.00 ± 15.56 mg/kg.

The best removal of aluminum, with 99.59% and 98.86%, from the pyrolysis oil was observed at low flow rates (0.05 mL/min) at 500 °C and 700 °C, respectively.

The results for the concentration of phosphorus showed similar behavior. The best removal efficiency of 2.00 ± 0.12 mg/kg (93.80%) was reported at 500 °C and 0.05 mL/min.

The bromine content was below 1.1 mg/kg at 500 °C, 0.05 mL/min and 600 °C, 0.25 mL/min. In these cases, the concentration of bromine was reduced below the limit of detection (LOD) of 1.10 mg/kg set by calibration of the ICP-OES analysis device. Considering the LOD as an actual concentration, the removal efficiency was 99.44% for both conditions.

The level of lead in the reforming product was reduced below the LOD of 0.09 mg/kg in the case of experiments at 500 °C, 0.05 mL/min and 600 °C, 0.25 mL/min. The ICP-OES analyses of lead concentrations comprised very high deviations of more than ±50% for experiments at temperatures of 700 °C, as well as at 500 °C, 0.45 mL/min, and are therefore not evaluated in detail.

The above-described results are shown in Figure 7 and Figure 8.

## 4. Discussion

### 4.1. Aromatization

Regarding the statistical evaluation conducted within this study, it should be noted that the linear model used is meant to reveal reliable data on the general dependency on the investigated parameters rather than provide a dependable forecast model for process design at higher scales. In future studies, the model can be expanded by additional data points to optimize crucial parameters.

It was found that the pyrolysis char contains several elements that can be present in an active form to promote reactions like cracking, adsorption, and reforming to aromatic compounds. For comparison, selected species were converted into their oxide form. The corresponding proportions are shown in Table 5 in contrast to contents in biochar and char from the pyrolysis of waste from electric and electronic equipment. Physical properties of the MPO323 pyrolysis char like pore volume, pore size, and surface area were similar to pyrolysis chars used as catalytically active compounds by other researchers.

Compared to biochar and electronic waste char, the char produced in the present work contained higher amounts of aluminum and calcium that can lead to good decontamination performance for several elements. Calcium might also promote the formation of aromatic hydrocarbons [38]. In contrast to the chars produced in [22], zinc is present in MPO323 pyrolysis char. The presence of zinc is known to have a promotive effect on the formation of aromatic hydrocarbon and dehydrocyclization by increasing the amount of Lewis acidic sites [34]. The highest proportion of BTEX was obtained at 600 °C, which aligns with the work in [39]. Considering the semi-quantitative GC/MS method used in this work, pyrolysis and subsequent auto-catalytic reforming yielded more MAH and TAH at 600 °C than a comparable process with waste tires as feedstock and zinc-loaded tire-derived char [39]. On the other hand, as proposed by Sun et al., the presence of CaO and iron species can inhibit the formation of PAH [33]. On the contrary, this investigation found excessive formation of PAH at higher temperatures. Hence, it can be assumed that Ca is not present in the form of an oxide, is not sufficiently accessible at the surface of the pyrolysis char or is not preserving the inhibition at very high temperatures. Generally, the formation of PAH at higher temperatures agrees with numerous other studies [40,41,42,43,44,45,46].

Sewage sludge-derived pyrolysis char produced higher liquid yields together with more MAH and particularly xylene from a comparable feedstock composition [31]. In contrast to this study, the feedstock was a model mixture of virgin PE, PP, and PS.

A statistically significant dependency with regard to the mass balance was verified for temperature and flow rate (*p*-values of <0.001 and <0.011, respectively; R-squared value of 0.959). A linear model for the prediction of the liquid yield is illustrated in Figure 9.

The proportion of AH in the pyrolysis oil was reduced in all reforming experiments. This followed the same trend as for the liquid yield—a lower proportion of AH was obtained with increasing temperature and decreasing flow rate. Statistical evaluation revealed that the formation of AH is evidently dependent on temperature (*p*-value of <0.001, R-squared value of 0.961). A plot of the derived model is shown in Figure 10.

Due to more complex behavior of the formation of BTEX comprising a supposed maximum between 500 °C and 700 °C, the selected model does not deliver any significance. Additional factorial points are necessary to validate statistical trends regarding the investigated parameters.

From the GC/MS peaks shown in Table 6, it can be assumed that some reforming reactions occurred during the process. Considering the liquid yield of 53.5% of the corresponding experiment, a slight increase in benzene yields can be observed. This is most likely due to dealkylation of certain aromatic hydrocarbons of higher molecular weight or due to dehydrocyclization of aliphatic compounds. Furthermore, the proportion of ethylbenzene did not increase along with the reduction of the amount of liquid product. Hence, reforming reactions of ethylbenzene to either benzene or other aromatic hydrocarbons can be assumed. Compared to the initial proportion, additional *o*-xylene was formed, which substantiates the assumption that several reforming reactions took place. Additionally, some branched aromatics were detected that were not present in the initial pyrolysis oil, as shown in Table 6. This leads to the overall conclusion that the investigated pyrolysis char does promote certain reforming reactions, possibly also aromatization of aliphatic compounds. Although, it can be observed that cracking of aliphatic compounds exceeded potential dehydrocyclization and reforming reactions, leading to diminished liquid yields, at temperatures higher than 500 °C.

The formation of PAH was observed from temperatures of 600 °C (4.71 ± 0.18 area%) up to 25.75 ± 5.73 area% at 700 °C and 0.05 mL/min, increasing with lower flow rates. Statistically, the dependency on all factors (temperature, flow rate, and temperature/flow rate cross interaction) was evident (*p*-values of <0.001, <0.04, and <0.4, respectively; R-squared value of 0.906). The model is visualized in Figure 11.

Competing trends regarding the liquid yield and the content of aromatics were observed in this study. With regard to economic factors of potential technical applications of this process, e.g., methods for the modification of the pyrolysis char can be explored in future studies to balance those competing outcomes. This can include the activation of the pyrolysis char with agents like sodium hydroxide, phosphoric acid, or simple calcination and subsequent reaction steps or fixed beds with technical catalysts like zeolites that are less susceptible to deactivation after the initial treatment with pyrolysis char. Additional analysis methods like SEM-EDX could provide insights regarding the bond type of different elements present in the pyrolysis char in future studies. This could allow more detailed insights into the reactions that may take place.

### 4.2. Decontamination of Pyrolysis Oils

One aim of this work was to investigate the char from MPO323 pyrolysis for its capability as a decontamination agent or, in other words, for its capability to reduce the concentrations of chlorine and bromine, as well as sulfur, iron, phosphorus, lead, and aluminum in the liquid pyrolysis product from MPO323 pyrolysis. With the char from MPO323 pyrolysis, pyrolysis oils of substantially reduced halogen contents can be produced. The raw oil from MPO323 pyrolysis exhibited average chlorine and bromine concentrations of 42,887 mg/kg and 195 mg/kg, respectively. The chlorine content in pyrolysis oil derived from MPO323 was exceptionally high. This leads to the assumption that the PVC content was higher than usual in the investigated waste sample. Calibration standards for ICP-OES analysis were measured for up to 10,000 mg/kg. A calibration curve for higher concentrations was extrapolated. Supplementary control standards with trichlorobenzene validated the correct manner of the obtained results, with less than 4% deviation at 50,000 mg/kg. The chlorine concentrations of the pyrolysis oil measured by ICP-OES might be higher than the real values, which could be subject to highly volatile chlorine-containing compounds like hydrogen chloride, which could be excessively released by the nebulizer and finally in the spray chamber of the ICP-OES device. The authors expect potential overestimation of chlorine content mostly for the initial raw pyrolysis oil samples prior to conducted reforming experiments. Additional analysis by, e.g., X-ray fluorescence analysis (XRF) and additional sample preparation could be applied to the pyrolysis oils to dissolve potentially present volatile chlorine-containing compounds prior to analysis in future studies.

Among the experiments where the char from MPO323 pyrolysis was applied, the highest average chlorine and bromine contents, at 4919 mg/kg and 17 mg/kg, respectively, were observed in the experiment with a reforming temperature of 700 °C and a flow rate of 0.45 mL/min. The lowest average chlorine content of 141 mg/kg and the lowest average bromine content of below 1 mg/kg were achieved in the experiment with a reforming temperature of 500 °C and a flow rate of 0.05 mL/min, constituting a reduction in chlorine content by 89% and in bromine content by over 99%. Since applying the char from MPO323 pyrolysis leads to substantially lower total halogen contents in the pyrolysis oils, it is an effective dehalogenation agent.

A similarly high reduction of the chlorine content in the pyrolysis oil by 88% to 58 mg/kg was achieved by Cho et al. using CaO, however, at reforming temperatures higher than 700 °C [20]. In the case of bromine, similarly good results were achieved by Brebu et al. Using FeOOH as sorbent in the pyrolysis of a mixture of PE, PP, PS, and ABS with bromine flame retardant and PVC led to a reduction of the bromine content of the pyrolysis oil by 95% to 104 mg/kg [47]. The investigated pyrolysis char was proven to be very efficient at removing chlorine at a temperature of 500 °C. Most likely, iron and calcium species that are present in high amounts (see Table 4) cause the release of chlorine from the pyrolysis oil, as proposed by Hubácek et al. [48]. The increase of chlorine in the liquid product aligns with their assumption that chlorides are released from sorbents at higher temperatures [48]. Sodium species with even higher removal efficiency towards chlorine could also promote good dechlorination performance, as proven by Jeong et al. [49].

This comparison of the current work’s results with results reported in literature also indicates that the effectivity of the char from MPO323 pyrolysis for reduction of the halogen content in the pyrolysis oil is comparable or even slightly superior to conventional, frequently investigated dehalogenation agents. This highlights the high potential of pyrolysis char for the inexpensive production of pyrolysis oil with low halogen content. Due to high deviation of the chlorine content in the product after experiments at 700 °C, only the influence of the flow rate was significant (*p*-value < 0.05). The authors assume that the removal efficiency is dependent on the temperature. Although, statistical significance needs to be validated by additional repetitions or an expanded factorial design of experiments. The same applies to the reduction of bromine in the product (*p*-value < 0.1). Up to now, most works by other researchers regarding the debromination of plastic pyrolysis oils have investigated waste electric and electronic equipment and construction materials [7]. Mixtures with a high content of polyolefins or comparable feedstocks have rarely been examined with reference to bromine. Bhaskar and colleagues showed in several publications that a calcium carbonate- and iron-based catalyst showed good performance to fully remove bromine and chlorine from pyrolysis oils derived from mixed plastics [50,51,52]. Although the investigated plastic mixtures were either model mixtures or otherwise not fully comparable to the one of the present study, it can be assumed that calcium and iron species contained in the pyrolysis char increase the debromination performance.

In general, fixation mechanisms for chlorine and bromine are reported to be similar, due to the alike nature of these elements [7]. Chen et al. found that the capacity of bromine for fixation onto metal oxides can be described in the following sequence: NaO_2_ > Fe_2_O_3_ > Al_2_O_3_ [53]. They propose that increased temperatures can lead to the re-release of Br into the pyrolysis oil, which aligns with the results of this study, although higher temperatures were investigated. Altarawneh et al. proposed the following mechanism for the fixation of HBr onto different metal oxides [54].
2yHBr + M_x_O_y_ → xMBr_2y/x_ + yH_2_O (M: Na, Ca, Fe, Al, Ti)(2)

They further assumed that bromine fixation of brominated hydrocarbons takes place by either direct elimination or dissociative adsorption. The authors suggest that similar mechanisms can be assumed for the fixation of bromine and chlorine regarding the present investigation. This leads to the hypothesis that the presence of sodium, calcium, iron, aluminum, and titanium (29.5 wt% in total) in the tested pyrolysis char have synergistic effects to remove chlorine and bromine from the pyrolysis oil. Additionally, the authors propose that dissociative adsorbed compounds can be released again due to desorption of the same at higher temperatures, which has been observed for both chlorine and bromine at experiments at 700 °C.

A linear model for the content of sulfur is shown in Figure 12. The influence of temperature was statistically significant (*p*-value: <0.001, R-squared value: 0.911). A statistically proven dependency on the flow rate was not confirmed from the generated data.

The concentration of phosphorus in the liquid product was dependent on both temperature and flow rate, as well as on their cross interaction (*p*-values: <0.09, <0.005, <0.035, respectively) The regression of the model was lacking statistical evidence (R-squared value: 0.833). The model is visualized in Figure 13.

The presence of numerous different species in the pyrolysis oil and the pyrolysis char do not allow to prove specific mechanisms proposed in the literature. On the other hand, the results indicate that the pyrolysis char has the potential to be utilized as an effective first upgrading agent that is cheap and available. The authors are of the opinion that the utilization of pyrolysis char as a post-treatment agent and potentially as an online treatment agent for pyrolysis oil upgrading enables relevant applicability in real-world scenarios, overcoming issues of technical adsorbents and catalysts being rapidly deactivated due to the high loads of contaminants in pyrolysis oils from MSPW sorting residues. To prove the applicability, post-treatment and online treatment tests should be conducted in larger-scale experiments. Experiments on the functional lifespan of char until less activity towards relevant decontamination performance is observed can underline the relevance of this technology in future investigations.

## 5. Conclusions

In this study, a process for the treatment of pyrolysis oil from municipal solid plastic waste (MSPW) using a recycling-based catalyst material was presented. The process aimed to improve the quality and stability of pyrolysis oil by removing contaminants such as heteroatoms and metals, increasing the aromaticity, and reducing the oxygen content. The effects of temperature and weight hourly space velocity (WHSV) on the product yields, composition, and properties were investigated.

In conclusion, the process was found to be suitable for the effective removal of different contaminants from pyrolysis oil and for increasing the content of aromatic compounds in the liquid product. However, the liquid yields were diminished by increasing temperature and WHSV due to additional cracking reactions. With increasing temperature and flow rate (correspondingly higher WHSV), a higher proportion of aromatic hydrocarbons was determined in the liquid product, and at 700 °C fully aromatic oils were produced. At temperatures higher than 600 °C, the formation of PAH was statistically evident, with excessive amounts at temperatures above 600 °C. The proportion of the most valuable compounds in the pyrolysis oil (BTEX) was maximized at 600 °C. It was shown that a high-quality mono-aromatic drop-in feedstock could be produced from MSPW pyrolysis oil. In addition, the contents of chlorine, aluminum, iron, phosphorus, and bromine were reduced by 99.7%, 99.6%, 98.1%, 93.8%, and 99.4%, respectively, by the presented method. The most favorable parameters for high decontamination performance were WHSV > 1 h^−1^ and temperatures lower than 600 °C. Further investigations should cover the effects of specific elements present in the pyrolysis char with regard to reforming and cracking reactions, as well as the removal of different contaminants. This could help to optimize the composition of alternative, recycling-based materials for refining purposes and ultimately enable processes that are less dependent on newly mined fossil resources.

## 6. Patents

Fraunhofer UMSICHT holds a patent to produce highly aromatic pyrolysis oil by post-treatment of pyrolysis vapors with the corresponding pyrolysis char as an active material [55]. The authors recently applied for another patent regarding decontamination of pyrolysis oils and other hydrocarbon-containing mixtures with the help of pyrolysis char as an active material. The application is directly linked to the results presented. The application process is pending.

## Figures and Tables

**Figure 1 polymers-16-02567-f001:**
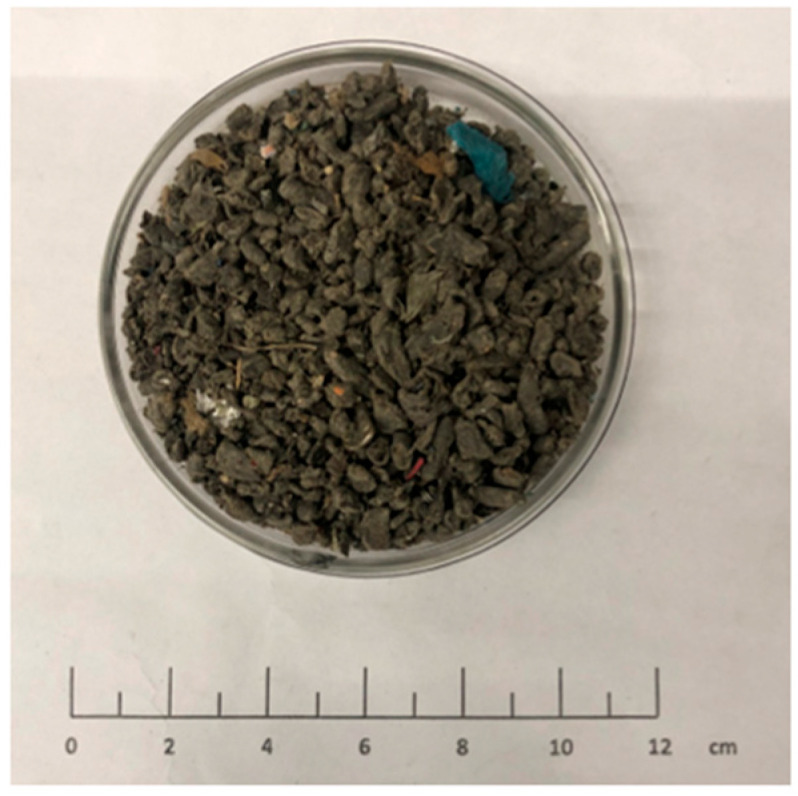
Mixed polyolefin plastic waste sorting residue (MPO323).

**Figure 2 polymers-16-02567-f002:**
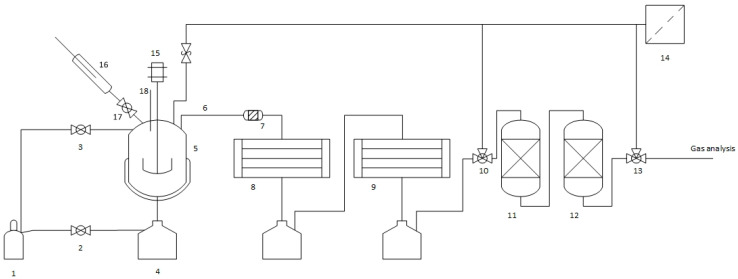
Process flow diagram of the pyrolysis pilot plant (thermo-chemical conversion process).

**Figure 3 polymers-16-02567-f003:**
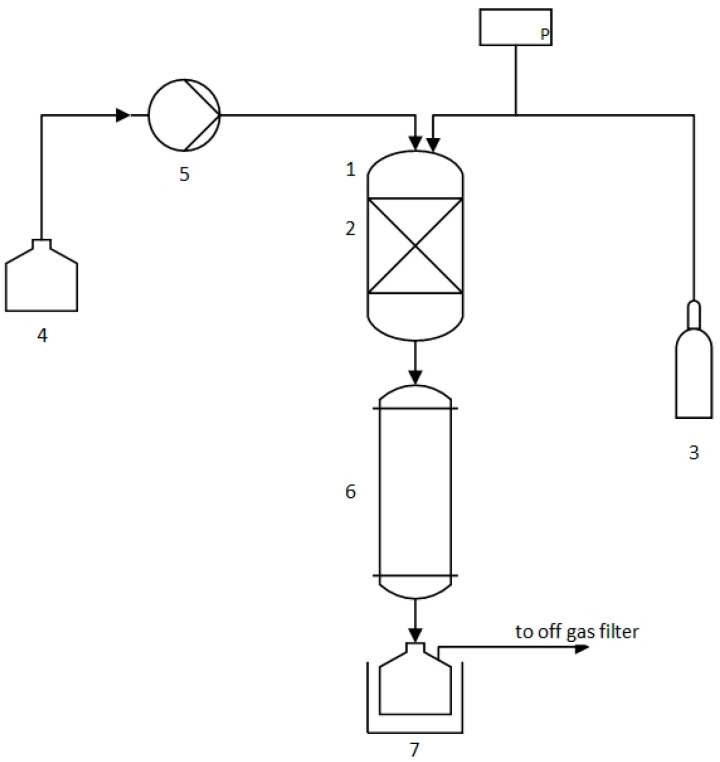
Process flow diagram of the laboratory set-up for auto-catalytic reforming.

**Figure 4 polymers-16-02567-f004:**
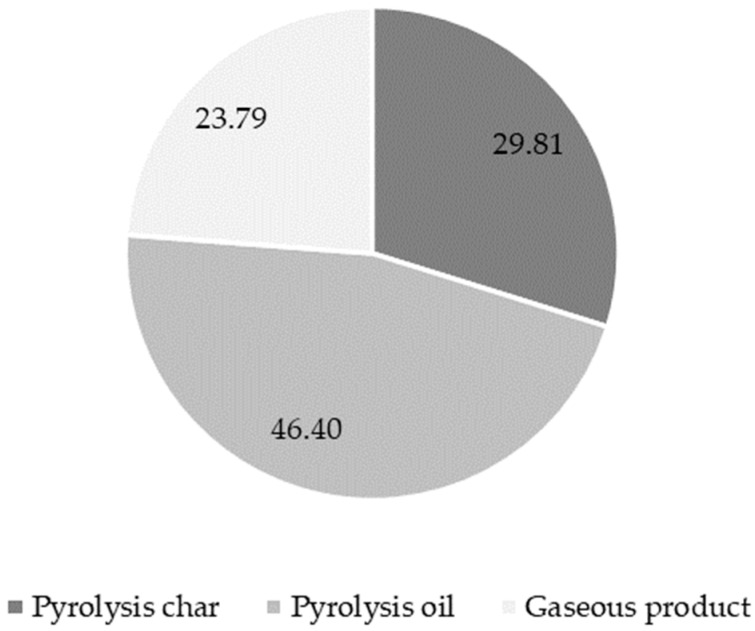
Mass balance of pilot scale pyrolysis experiments.

**Figure 5 polymers-16-02567-f005:**
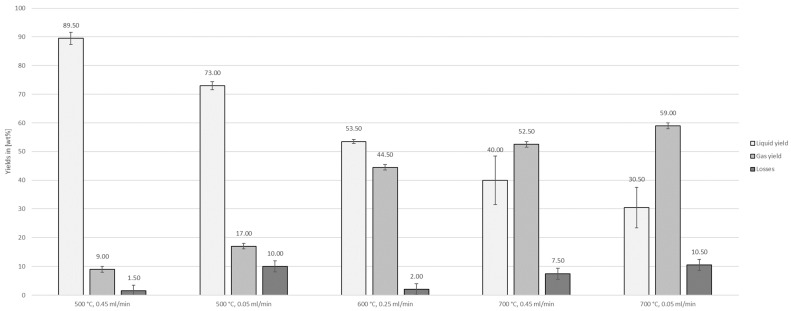
Liquid yields, gas yields, and losses of auto-catalytic reforming experiments.

**Figure 6 polymers-16-02567-f006:**
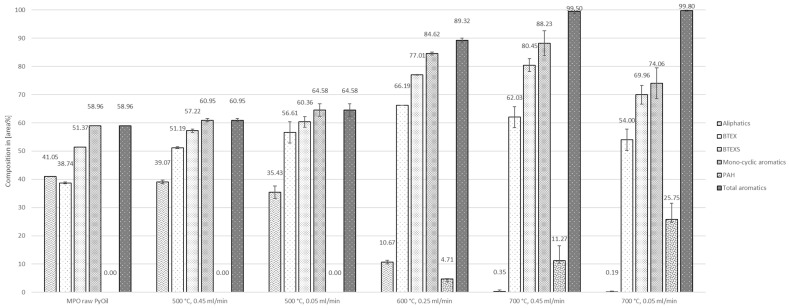
Composition of MPO323 pyrolysis oil and liquid products from auto-catalytic reforming experiments.

**Figure 7 polymers-16-02567-f007:**
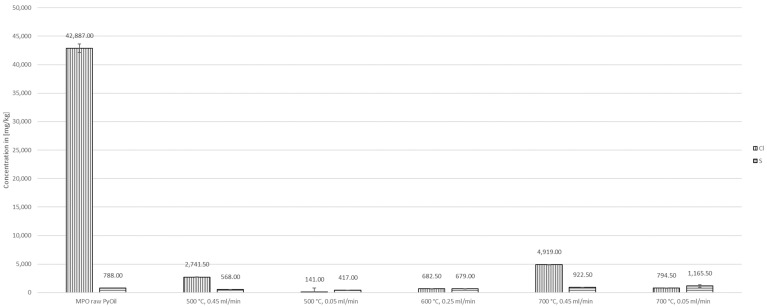
Concentrations of chlorine and sulfur in MPO323 pyrolysis oil and liquid products from auto-catalytic reforming experiments.

**Figure 8 polymers-16-02567-f008:**
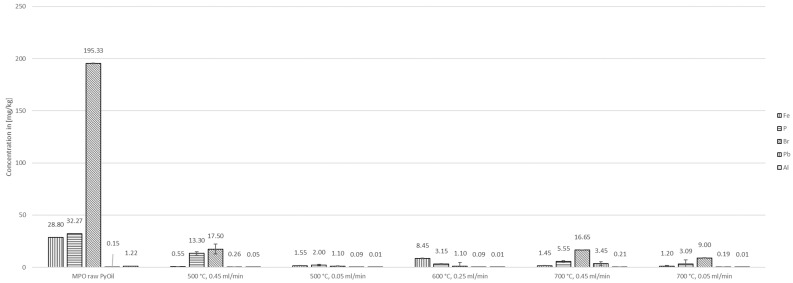
Concentrations of iron, phosphorus, bromine, lead, and aluminum in MPO323 pyrolysis oil and liquid products from auto-catalytic reforming experiments.

**Figure 9 polymers-16-02567-f009:**
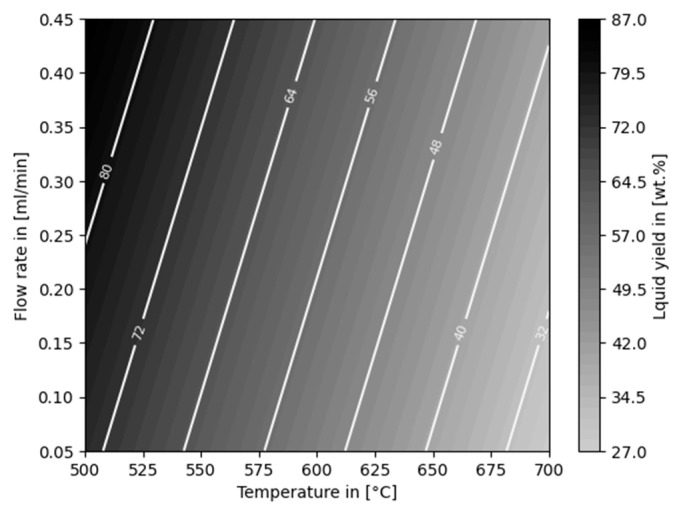
Linear model for liquid yield dependent on temperature and flow rate.

**Figure 10 polymers-16-02567-f010:**
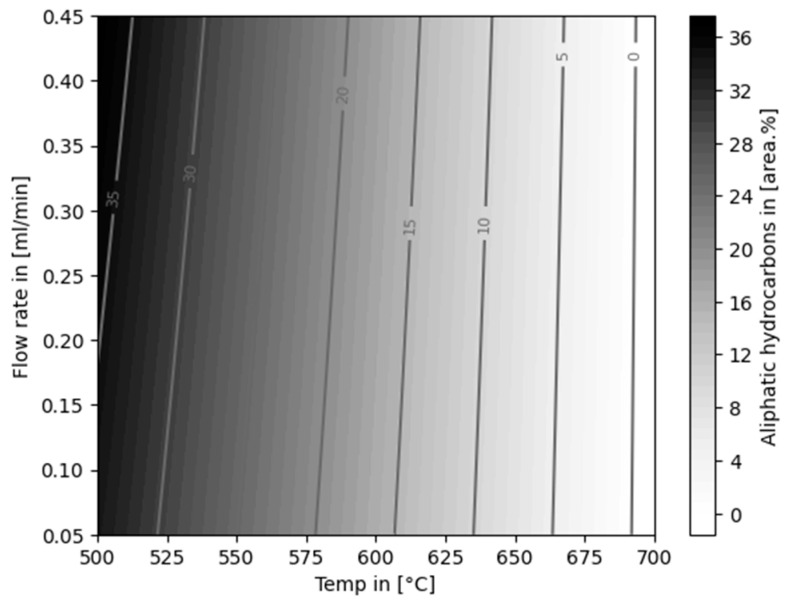
Linear model for the content of aliphatic hydrocarbons in the liquid product, dependent on temperature and flow rate.

**Figure 11 polymers-16-02567-f011:**
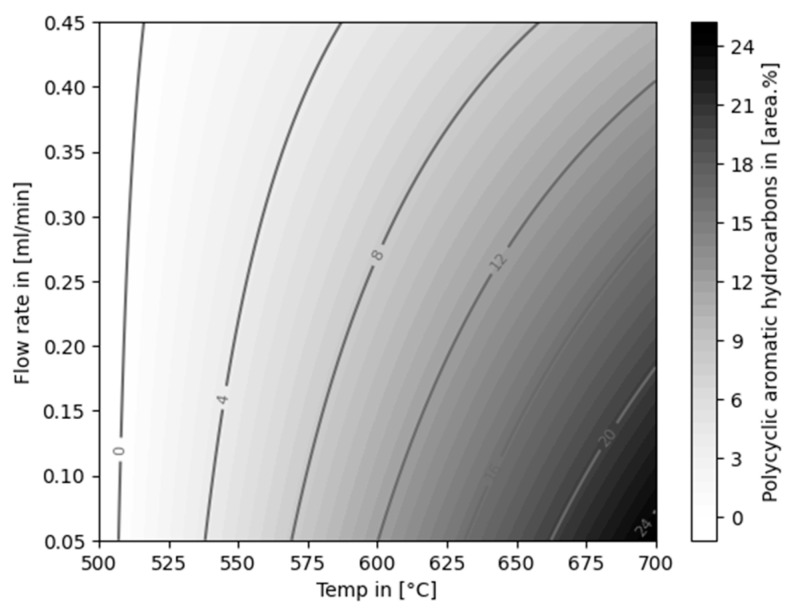
Linear model for the content of polycyclic aromatic hydrocarbons (PAHs) in the liquid product, dependent on temperature and flow rate.

**Figure 12 polymers-16-02567-f012:**
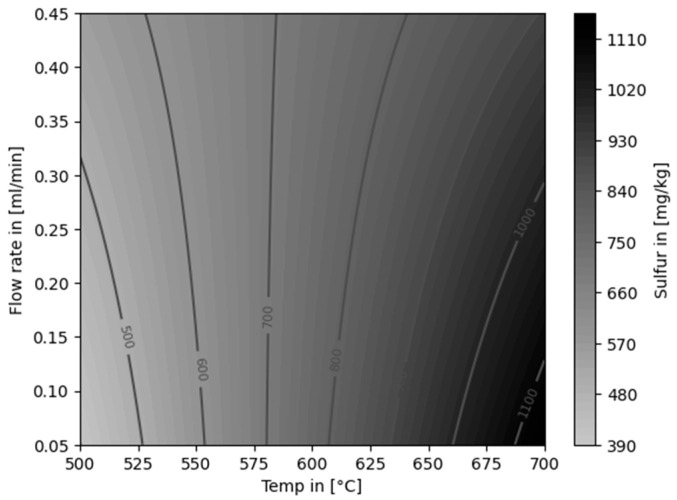
Linear model for the content of sulfur in the liquid product, dependent on temperature and flow rate.

**Figure 13 polymers-16-02567-f013:**
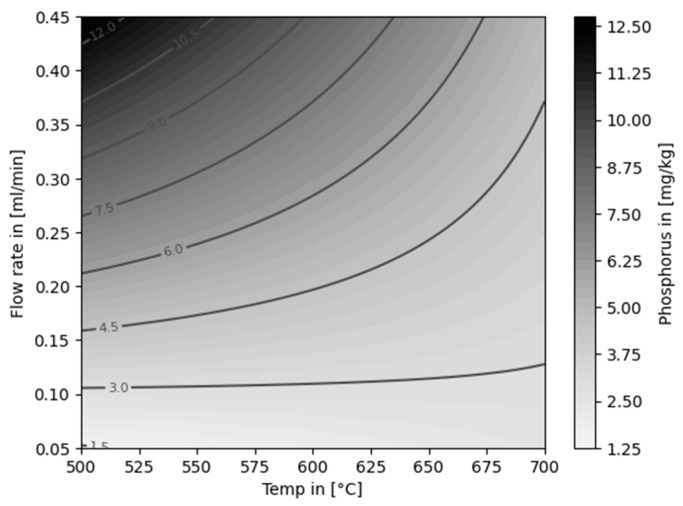
Linear model for the content of phosphorus in the liquid product, dependent on temperature and flow rate.

**Table 1 polymers-16-02567-t001:** Design of experiments.

Experiment	Catalytically Active Material	Temperature [°C]	Flow Rate [mL/min]	Feed Material
CR1_MPO_1	Raw MPO323 char	500	0.05	MPO323 oil
CR1_MPO_2	Raw MPO323 char	500	0.05	MPO323 oil
CR2_MPO_1	Raw MPO323 char	500	0.45	MPO323 oil
CR2_MPO_2	Raw MPO323 char	500	0.45	MPO323 oil
CR3_MPO_1	Raw MPO323 char	600	0.25	MPO323 oil
CR3_MPO_2	Raw MPO323 char	600	0.25	MPO323 oil
CR4_MPO_1	Raw MPO323 char	700	0.05	MPO323 oil
CR4_MPO_2	Raw MPO323 char	700	0.05	MPO323 oil
CR5_MPO_1	Raw MPO323 char	700	0.45	MPO323 oil
CR5_MPO_2	Raw MPO323 char	700	0.45	MPO323 oil

**Table 2 polymers-16-02567-t002:** GC/MS peak table of conventional MPO323 pyrolysis oil produced at 520 °C.

Area%	Ret. Time	Name
19.10	7.468	Ethylbenzene
12.82	8.111	Styrene
11.10	5.486	Toluene
8.73	6.994	2,4-Dimethyl-1-heptene
6.87	3.591	Benzene
3.54	10.000	1-Decene
3.34	5.242	AH C8
3.19	8.055	AH C9
3.11	6.801	AH C8
2.98	15.020	AH C13
2.20	8.752	Benzene, (1-methylethyl)-
2.09	11.775	1-Undecene
2.06	5.957	1-Octene
1.99	9.574	AH C10
1.88	14.906	1-Tridecene
1.69	13.398	1-Dodecene
1.64	4.010	1-Heptene
1.62	9.893	Alpha-Methylstyrene
1.61	7.665	Xylene
1.59	15.267	AH C13
1.59	16.317	1-Tetradecene
1.47	17.646	1-Pentadecene
1.31	11.506	AH C11
1.27	9.650	Mesitylene
1.22	5.435	AH C8

**Table 3 polymers-16-02567-t003:** ICP-OES analysis of MPO323 pyrolysis oil.

Species	Concentration [ppm]	Standard Deviation [ppm]
Cl	42,887.00	747.24
S	788.00	3.46
Br	195.33	3.51
P	32.27	0.15
Fe	28.80	4.92
Al	1.22	0.43
Pb	0.15	0.06

**Table 4 polymers-16-02567-t004:** Composition of MPO323 pyrolysis char from the ultimate analysis and the ICP-OES analysis *.

Species	Concentration [wt%]	Species	Concentration [wt%]
Al	11.00	Mn	0.05
Ba	0.73	N	0.67
C	63.66	Na	1.60
Ca	12.00	Ni	0.06
Cu	2.10	P	0.34
Cr	0.06	Pb	0.05
Fe	1.80	S	0.14
H	3.14	Si	8.30
K	0.67	Ti	3.10
Mg	1.20	Zn	2.70

* Sum differs from 100% due to the use of different analytical methods.

**Table 5 polymers-16-02567-t005:** Comparison of elements converted to oxide form in pyrolysis char from MPO323 and other sources.

Name, Property	Unit	MPO323 Char	Biochar [22]	Electronic Waste Char [22]	Fe/Biochar [22]	Fe/Electronic Char [22]
SiO_2_	[wt%]	7.30	10.88	0.676	33.3	2.28
Al_2_O_3_	[wt%]	9.30	1.28	0.165	5.46	0.627
CaO	[wt%]	23.0	1.28	1.89	5.13	7.55
Fe_2_O_3_	[wt%]	2.20	1.12	1.21	17.2	8.92
TiO_2_	[wt%]	2.50	0.06	0	0.236	2.29
Na_2_O	[wt%]	0.93	0.82	0.112	3.33	1.5
MgO	[wt%]	0.80	0.48	0.421	2.03	0
ZnO	[wt%]	1.8	-	-	-	-
BET	[m^2^/g]	5.3	4.2	4.5	52.4	10.8
Pore volume	[cm^2^/g]	0.013	0.008	0.006	0.055	0.02
Pore size	[nm]	10.2	7.65	5.43	4.2	7.34

**Table 6 polymers-16-02567-t006:** Monocyclic aromatic hydrocarbons in the initial MPO323 pyrolysis oil and the auto-catalytically reformed product.

Initial Pyrolysis Oil		Auto-Catalytically Reformed at 600 °C, 0.25 mL/min	
Area%	Name	Area%	Name
6.87	Benzene	14.63	Benzene
11.1	Toluene	22.17	Toluene
19.1	Ethylbenzene	23.69	Ethylbenzene
1.61	Xylene	6.08	Xylene
12.82	Styrene	10.52	Styrene
2.2	Benzene, (1-methylethyl)-	1.73	Benzene, (1-methylethyl)-
		0.88	Benzene, 1-ethyl-3-methyl-
		0.68	Benzene, 1-ethyl-2-methyl-
		0.99	Benzene, 1,2,3-trimethyl-
1.62	Alpha-Methylstyrene	1.23	Alpha-Methylstyrene
1.27	Mesitylene	2.37	Mesitylene

## Data Availability

The original contributions presented in the study are included in the article, further inquiries can be directed to the corresponding author.

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
