# Peer review of "Chemical Recycling of Mixed Polyolefin Post-Consumer Plastic Waste Sorting Residues (MPO323)—Auto-Catalytic Reforming and Decontamination with Pyrolysis Char as an Active Material"

_polymers, 2024, doi:10.3390/polym16182567_

Round 1

Reviewer 1 Report

Comments and Suggestions for Authors

In this research, a method for treating pyrolysis oil derived from municipal solid plastic waste utilizing a recycling-based catalyst was introduced. The goal of this process was to enhance the quality and stability of the pyrolysis oil by eliminating contaminants such as heteroatoms and metals, increasing aromatic content, and lowering oxygen levels. The impact of temperature and weight hourly space velocity on the product yields, composition, and characteristics was examined.

The study is engaging and addresses an important topic. Some revisions are suggested prior to making a final decision.

1. Line 171 “Real municipal solid waste (MSPW)”, it seems that MSPW is the abbreviation of the municipal solid plastic waste not the municipal solid waste. Please check and revise.

2. Line 241-242 “To fit a model by using 241 a response surface model (RSM) and ordinary least square regression (OLS).” This sentence is incomplete and does not have verb. Please revise it.

3. Given the observed trends of decreasing liquid yield with increasing temperature and decreasing flow rate, how do you propose to optimize the pyrolysis oil reforming process to maximize both the yield and the desired aromatic compound content? Are there any alternative methods or modifications that could be explored to balance these competing outcomes?

4. How do you intend to address the potential overestimation of chlorine concentrations due to the presence of volatile chlorine-containing compounds like hydrogen chloride in your analysis? What measures could be implemented to improve the accuracy of chlorine quantification in future studies?

5. Including a comprehensive analysis of how the presence of chlorine and other elements in the pyrolysis char influences its catalytic properties could strengthen the relevance and applicability of your findings to real-world scenarios.

6. Considering the significant reduction of halogen content achieved with the char from MPO323 pyrolysis, what specific mechanisms do you propose are responsible for this dehalogenation effect at varying temperatures and flow rates? Additionally, how do you account for the discrepancies in chlorine reduction efficiency when comparing your results to those from other studies using different sorbents?

7. To enhance the quality of the article, consider including a more detailed discussion on the chemical mechanisms at play during the reforming process. This could involve exploring how the interactions between pyrolysis char and pyrolysis oil at varying temperatures and flow rates influence the formation of specific compounds, which would provide a deeper understanding of the underlying principles and guide future research in optimizing this process. Additionally, incorporating a comparison with existing literature on similar processes could strengthen your findings and contextualize your results within the broader field.

Author Response

Dear Sir or Madam,

Thank you very much for your effort to review our draft. We are pleased to see that you consider the research design, the methods used and the presentation of the results to be appropriate.

Please note that we considered your comments as follows:

Comment 1

Line 171 “Real municipal solid waste (MSPW)”, it seems that MSPW is the abbreviation of the municipal solid plastic waste not the municipal solid waste. Please check and revised.

Answer 1

The abbreviation has been revised.

Comment 2

Line 241-242 “To fit a model by using 241 a response surface model (RSM) and ordinary least square regression (OLS).” This sentence is incomplete and does not have verb. Please revise it.

Answer 2

The sentence has been revised.

Comment 3

Given the observed trends of decreasing liquid yield with increasing temperature and decreasing flow rate, how do you propose to optimize the pyrolysis oil reforming process to maximize both the yield and the desired aromatic compound content? Are there any alternative methods or modifications that could be explored to balance these competing outcomes?

Answer 3

The discussion has been extended with respect to your comment (L472-481).

Comment 4

How do you intend to address the potential overestimation of chlorine concentrations due to the presence of volatile chlorine-containing compounds like hydrogen chloride in your analysis? What measures could be implemented to improve the accuracy of chlorine quantification in future studies?

Answer 4

The corresponding paragraph in the discussion has been expanded and moved to section 4.2 (L489-503). The authors expect a potential overestimation of chlorine content only for the initial raw pyrolysis oil samples prior to conducted reforming experiments as a result of the limitations of the applied analytical method. Additional sample preparation could be applied to raw oils to dissolve volatile chlorine-containing compounds prior to analysis to overcome this effect. If so, this should likewise be applied for the resulting products to ensure comparability which, on the other hand, bears the risk of distorting those results since an additional treatment step is employed. Hence, the authors are of the opinion that the chosen method is suitable to prove that an excessive chlorine content is present in the raw pyrolysis oil and can be effectively decreased by introducing the corresponding pyrolysis char as an upgrading agent in this demonstrative study. Within this context it is crucial to obtain reliable and comparable analytical results for the upgraded product. In future studies, chlorine analysis could be supplemented by other analysis methods like XRF.

Comment 5 

Including a comprehensive analysis of how the presence of chlorine and other elements in the pyrolysis char influences its catalytic properties could strengthen the relevance and applicability of your findings to real-world scenarios.

Answer 5

The discussion has been expanded with respect to real world applications of this process (L573-584).

Comment 6

Considering the significant reduction of halogen content achieved with the char from MPO323 pyrolysis, what specific mechanisms do you propose are responsible for this dehalogenation effect at varying temperatures and flow rates? Additionally, how do you account for the discrepancies in chlorine reduction efficiency when comparing your results to those from other studies using different sorbents?

Answer 6

Considering that the reaction system of pyrolysis oil with its corresponding pyrolysis char is highly complex due to the presence of an extremely high number of reactants that cannot all be determined regarding their type (e.g. oxide, metal, etc) and resulting cross interactions and competing reactions, the authors believe that a proposition of specific reaction mechanisms would be quite audacious. Nevertheless, a discussion on possible fundamental mechanisms has been added (L544-559).

Comment 7

To enhance the quality of the article, consider including a more detailed discussion on the chemical mechanisms at play during the reforming process. This could involve exploring how the interactions between pyrolysis char and pyrolysis oil at varying temperatures and flow rates influence the formation of specific compounds, which would provide a deeper understanding of the underlying principles and guide future research in optimizing this process. Additionally, incorporating a comparison with existing literature on similar processes could strengthen your findings and contextualize your results within the broader field.

Answer 7

Regarding your comment, the discussion has been updated discussing possible reaction mechanisms and contextualizing the findings in a broader field and the applicability of the process in real world scenarios.

We highly appreciate your comments which have contributed to enhance the quality of the draft.

With best regards,

The authors of the study

Reviewer 2 Report

Comments and Suggestions for Authors

Comments

My comment for the manuscript entitled, ‘Chemical Recycling of Mixed Polyolefin Post-Consumer Plastic Waste Sorting Residues (MPO323) – Auto-Catalytic Reforming and Decontamination with Pyrolysis Char as an Active Material’ is given below, 

The manuscript deals with the chemical recycling process of polyolefin plastic waste and the extraction of pyrolysis oil with the essential components. The manuscript will be more suitable for other journals related to chemical processes.

Author Response

Dear Sir or Madam,

Thank you very much for your effort to review our draft. We are pleased to see that you consider the research design, the methods used and the presentation of the results to be appropriate.

Comment 1:

The manuscript deals with the chemical recycling process of polyolefin plastic waste and the extraction of pyrolysis oil with the essential components. The manuscript will be more suitable for other journals related to chemical processes.

Response 1:

We agree that the publication does not directly investigate the production, modification or direct application or virgin polymers. Although, we are planning to contribute the presented work to MPDI Polymers’s section ‘Circular and Green Polymer Science’. The scope of this section is defined as follows: ‘Design and development of green polymer systems; mechanical and chemical recycling and energy recovery of polymer and biopolymer-based systems; polymer synthesis using renewable resources and employing green methods for polymer synthesis.’ Since research in the field of chemical recycling is one of the explicitl mentioned, we believe that this contribution fits well within the scope of the journal.

With best regards,

The authors of the study

Reviewer 3 Report

Comments and Suggestions for Authors

The authors thoroughly investigated the recycling process of different contaminations. Various yields were measured and compared. This information attracts the interest of some readers. This is worth publishing. If you can, please revise the following point. Linear model was used in the present manuscript. I recommend the authors commenting on the plausibility of this model by comparing this with other models.

Author Response

Dear Sir or Madam,

Thank you very much for your effort to review our draft. We are pleased to see that you consider the research design, the methods used and the presentation of the results to be appropriate.

Comment 1:

If you can, please revise the following point. Linear model was used in the present manuscript. I recommend the authors commenting on the plausibility of this model by comparing this with other models.

Response 1:

We added a comment in the discussion on the reliability and the scope of using a linear model (L397-401). A dedicated comparison to other models cannot be provided as no suitable data was found to be publicly available.

We highly appreciate your comments which have contributed to enhance the quality of the draft.

With best regards,

The authors of the study

Round 2

Reviewer 2 Report

Comments and Suggestions for Authors

The given comments are rectified